# Differential Expression and Function of SVIP in Breast Cancer Cell Lines and In Silico Analysis of Its Expression and Prognostic Potential in Human Breast Cancer

**DOI:** 10.3390/cells12101362

**Published:** 2023-05-11

**Authors:** Esra Atalay Şahar, Petek Ballar Kirmizibayrak

**Affiliations:** 1Department of Biotechnology, Graduate School of Natural and Applied Sciences, Ege University, Izmir 35100, Turkey; 2Department of Biochemistry, Faculty of Pharmacy, Ege University, Izmir 35100, Turkey

**Keywords:** SVIP, endoplasmic reticulum-associated degradation, breast cancer, in silico analysis

## Abstract

The heterogeneity of cancer strongly suggests the need to explore additional pathways to target. As cancer cells have increased proteotoxic stress, targeting proteotoxic stress-related pathways such as endoplasmic reticulum stress is attracting attention as a new anticancer treatment. One of the downstream responses to endoplasmic reticulum stress is endoplasmic reticulum-associated degradation (ERAD), a major degradation pathway that facilitates proteasome-dependent degradation of unfolded or misfolded proteins. Recently, SVIP (small VCP/97-interacting protein), an endogenous ERAD inhibitor, has been implicated in cancer progression, especially in glioma, prostate, and head and neck cancers. Here, the data of several RNA-sequencing (RNA-seq) and gene array studies were combined to evaluate the *SVIP* gene expression analysis on a variety of cancers, with a particular focus on breast cancer. The mRNA level of SVIP was found to be significantly higher in primary breast tumors and correlated well with its promoter methylation status and genetic alterations. Strikingly, the SVIP protein level was found to be low despite increased mRNA levels in breast tumors compared to normal tissues. On the other hand, the immunoblotting analysis showed that the expression of SVIP protein was significantly higher in breast cancer cell lines compared to non-tumorigenic epithelial cell lines, while most of the key proteins of gp78-mediated ERAD did not exhibit such an expression pattern, except for Hrd1. Silencing of SVIP enhanced the proliferation of p53 wt MCF-7 and ZR-75-1 cells but not p53 mutant T47D and SK-BR-3 cells; however, it increased the migration ability of both types of cell lines. Importantly, our data suggest that SVIP may increase p53 protein levels in MCF7 cells by inhibiting Hrd1-mediated p53 degradation. Overall, our data reveal the differential expression and function of SVIP on breast cancer cell lines together with in silico data analysis.

## 1. Introduction

Breast cancer is the most commonly diagnosed cancer in women. The incidence rate of breast cancer in women surpassed lung cancer, with 2.3 million new cases (11.7% of total cases) recorded in 2020, making it the most prevalent cancer type worldwide [1,2]. Even though great efforts have been made to identify new therapeutic strategies or potential biomarkers by different approaches, breast cancer remains the leading cause of cancer-related mortality in women. Given its heterogeneity with high relapse and drug resistance rates, identifying new prognostic biomarkers or therapeutic targets is still urgently needed in breast cancer research and treatment.

The endoplasmic reticulum is a central organelle that is involved in the maintenance of cellular homeostasis and the balance between health and disease. It regulates proper protein folding, modification, and quality control [3,4]. Multiple adverse external and intracellular factors, such as hypoxia, nutrient deprivation, cancer, obesity, neurodegeneration, oxidative stress, and viral infections, can disrupt endoplasmic reticulum homeostasis. When endoplasmic reticulum homeostasis is disrupted, and the protein folding capacity is exceeded, cells trigger a condition defined as “endoplasmic reticulum stress” [5]. The cellular response to endoplasmic reticulum stress is the unfolded protein response (UPR), a well-established signaling pathway. In order to reestablish the normal functioning of the endoplasmic reticulum and proteostasis, UPR activates several strategies in parallel and in series. These strategies include the prevention of protein aggregation via enhancing endoplasmic reticulum chaperon expressions, attenuation of global protein translation, and promoting degradation of misfolded and unfolded proteins from the endoplasmic reticulum using the endoplasmic reticulum-associated degradation (ERAD) pathway [6,7,8].

ERAD is a multi-component and complex system that involves the recognition of misfolded proteins by the appropriate luminal chaperones, retrotranslocation into the cytosol, polyubiquitination, and degradation by 26S proteasome [8,9]. The AAA-ATPase Valosin-containing protein (VCP, also known as p97) is the key protein functioning in the retrotranslocation of ERAD substrates across the endoplasmic reticulum membrane into the cytosol. Besides ERAD, p97/VCP also participates in autophagy and mitophagy [10,11]. As it has also been revealed that p97/VCP is involved in cancer cell reprogramming [12], the regulation of p97/VCP activity is crucial for cellular homeostasis. There are several identified functional partners of p97/VCP; three of them, namely E3 ubiquitin ligase gp78, small VCP/p97-interacting protein (SVIP), and VCP-interacting membrane protein (VIMP, also known as selenoprotein S) possess VCP-interacting motif (VIM) in their structure. While gp78 and VIMP promote the ERAD process, SVIP acts as an endogenous ERAD inhibitor [13,14,15].

Recent studies have suggested that SVIP may be involved in cancer progression. Firstly, SVIP levels were downregulated after androgen treatment in prostate cancer cells, while other components of ERAD machinery were upregulated [16], which was found to be positively related to prostate tumorigenesis. Similarly, SVIP was reported to be downregulated by androgen in the glioma cells and suggested as a new target for new for p53wt gliomas [17]. In a study that aimed to identify putative genetic and epigenetic changes in the p97/VCP-mediated ERAD pathway in human tumors, SVIP promoter CpG island was found to be methylated in 50% (19 of 38) of head and neck cancer cell lines. Additionally, SVIP undergoes DNA hypermethylation also in esophageal (8 of 35; 23%) and cervical (3 of 14; 21%) cancers and hematological malignancies (22 of 154; 14%), particularly B cell lymphoma (15 of 45; 33%). Apart from these cancer types, the SVIP promoter CpG island was most often found to be unmethylated in the other cancer types [18]. Moreover, a comparison of SVIP expression in different prostate cell lines revealed that SVIP is highly expressed in androgen-dependent prostate cancer cells (LNCAP, 22RV1) but not in androgen-independent cell lines (PC3, DU145) or non-tumorigenic prostate cell lines, including normal prostate epithelial cell lines (RWPE1) and benign prostatic hyperplasia epithelial cell lines (BPH1) [16].

Considering all this data suggesting that SVIP may play a role in tumor progression, we have first performed in silico analysis of SVIP expression on various cancers, with a particular focus on breast cancer. We examined the association between SVIP and breast cancer by performing comprehensive bioinformatics analysis on several large online databases and evaluated the expression profile in a variety of breast cancer cell lines. While SVIP mRNA expression was much higher in breast cancer tissue compared to control tissue, SVIP protein level was found to be significantly lower. SVIP silencing augmented the proliferation of p53 wt MCF-7 and ZR-75-1 cells, but not p53 mutant T47D and SK-BR-3 cells, and enhanced the migration ability of both types of cell lines. Our data revealing the differential expression and function of SVIP on breast cancer cell lines, together with in silico data, suggest that SVIP may have a tumor suppressor function in breast cancer progression.

## 2. Materials and Methods

### 2.1. Gene Expression Analysis

*SVIP* gene expression levels were analyzed through the TNMplot database (http://www.tnmplot.com, accessed on 29 March 2022) [19]. The Tumor Immune Estimation Resource (TIMER) database (https://timer.comp-genomics, accessed on 29 March 2022) [20] was used to verify the expression of SVIP in various cancers. TNMplot database (http://www.tnmplot.com, accessed on 29 March 2022) with RNA-sequence data was used to explore the SVIP gene expression in breast cancer tumors, normal, and metastatic tissues. TNMplot database provides gene array data from the Gene Expression Omnibus of the National Center for Biotechnology Information (NCBI-GEO), RNA-seq datasets from The Cancer Genome Atlas (TCGA), and also Therapeutically Applicable Research to Generate Effective Treatments (TARGET) and The Genotype-Tissue Expression (GTEx).

### 2.2. Promoter Methylation Analysis of SVIP

With data obtained from the TCGA project, the UALCAN database is an interactive web portal that enables gene expression analysis on about 20,500 protein-coding genes in 33 different tumor types [21]. In addition, this tool was also used to find the promoter methylation of SVIP in breast cancer. Furthermore, the correlation between methylation and expression was analyzed using cBioPortal. cBioPortal for Cancer Genomics (v3.7.27 http://www.cbioportal.org/, accessed on 13 April 2022) database. The portal contains data sets from published cancer studies, including the Cancer Cell Line Encyclopedia (CCLE) and the TCGA pipeline [22,23].

### 2.3. Genomic Analysis

cBioPortal for Cancer Genomics (v3.7.27, http://www.cbioportal.org/, accessed on 10 May 2022) is a powerful platform that is used for exploring, visualizing, and analyzing multidimensional cancer genomics data [22,23]. The genomics features of *SVIP* on breast cancer were analyzed via the cBioPortal database. The gene alteration frequency of *SVIP* using the TCGA and PanCancer Atlas from 1084 breast cancer patients was analyzed. The alterations included amplification, deep deletion, mRNA expression, and truncating mutation. Mutation details, copy number alterations (CNA), and genomic alteration of *SVIP* were also taken into consideration in different breast cancer types. CNA, mutation details, and mRNA expression were generated from RNA-seq (log2) data with default settings and plotted with mRNA expression data using the cBioPortal database. The mRNA expression z-score threshold was ± 2 between the unaltered and altered patients.

### 2.4. SVIP Gene Expression Based on Intrinsic Molecular Subtype and Clinicopathological Data Analysis

The Breast Cancer Gene-Expression Miner v4.8 (bc-GenExMiner v4.8), a mining tool, UALCAN, and the Gene Expression Profiling Interactive Analysis (GEPIA) (http://gepia.cancer-pku.cn/, accessed on 6 April 2022) databases were used evaluation of *SVIP* expression based on clinicopathological characteristics and intrinsic molecular subtype (PAM50 cancer subtypes) [21,24,25]. The clinicopathological parameters were as follows: estrogen receptor (ER) status, progesterone receptor (PR) status, HER2 receptor status, p53 status, pathological tumor stage, and nature of the tissue (healthy, tumor, and tumor-adjacent). All data were from TCGA and GTEx datasets.

SVIP protein expression was compared between tumor and normal samples in breast cancer using UALCAN database. UALCAN database was also introduced to determine the SVIP protein expression levels based on intrinsic molecular subtype in breast cancer (https://ualcan.path.uab.edu/, accessed on 17 April 2022).

### 2.5. Survival Analysis

The effects of *SVIP* expression on the survival of patients with breast cancer were performed using the Kaplan–Meier Plotter online survival analysis tool (https://kmplot.com/analysis/, accessed on 6 April 2022) [26]. This tool is used in a meta-analysis-based discovery and validation of survival biomarkers. The prognostic indicators evaluated include overall survival (OS) and distant-metastasis-free survival (DMFS). Additionally, the prognostic role of *SVIP* expression was also analyzed in the different subtypes (TNBC, HER-2, luminal A, luminal B) of breast cancer using the Kaplan–Meier Plotter online tool (https://kmplot.com/analysis/, accessed on 8 May 2022).

### 2.6. Cell Culture and Treatments

All cell lines were obtained from the American Type Culture Collection (ATCC, Manassas, VA, USA) and cultured according to instructions of the ATCC. MCF10A cells were grown in DMEM/F12 medium with 5% horse serum, 100 µg/mL EGF, 1 mg/mL hydrocortisone, 1 mg/mL cholera toxin, and 10 mg/mL insulin. MCF-7, SK-BR-3, and MDA-MB-231 cells were cultured in Dulbecco’s Modified Eagle’s High Glucose Medium (DMEM) supplemented with 10% fetal bovine serum (FBS). BT474 cells were grown in DMEM High Glucose supplemented with 10% FBS, 10 μg/mL insulin, and 1% MEM non-essential amino acid. The T47D cell line was grown in DMEM High Glucose supplemented with 10% FBS and 0.1% MEM non-essential amino acid. ZR-75-1 cell line was grown in Roswell Park Memorial Institute medium (RPMI) 1640 containing 10% FBS. All cell lines were grown as monolayers and were incubated at 37 °C with 95% humidified air and 5% CO_2_. Cells were checked regularly for mycoplasma contamination by PCR (MycoAlert™ Mycoplasma Detection Kit, Lonza, Basel, Switzerland).

For experiments using 17β-estradiol (E2) (Cayman Chemical, Ann Arbos, MI, USA), phenol red-free DMEM High glucose and Dextran-coated charcoal-treated FBS (DCC-FBS) (Biological Industries, Haemek, Israel) were used. MCF-7, T47D, and ZR-75-1 (3 × 10^5^ cells/well) were cultured in six-well plates at a confluency of 40%; after 1 day, the normal medium was replaced by phenol red-free DMEM High glucose supplemented with 5% DCT-FBS. Before E2 treatments, cells were cultured for 72 h in phenol red-free medium supplemented with 5% DCC-FBS. The medium was changed to 0.5% DCC-FBS at time 0 h, and E2 was added to a final concentration of 10 nM. E2 was then incubated for 30 m, 1 h, 2 h, 3 h, and 4 h.

Transfections were performed with Lipofectamin-2000 (Invitrogen, Carlsbad, CA, USA) following the manufacturer’s instructions. Silencer^®^ Negative Control siRNA (4611, Ambion, Austin, TX, USA), SVIP siRNA (AM16104, sense sequence: GACAAAAAGAGGCUGCAUC), and Hrd1 siRNA (AM16708, sense sequence: GGCCUUUGUCCUUGUCUUC) were ordered from Ambion.

### 2.7. Immunoblotting

Cells were harvested and lysed using RIPA buffer (1X PBS, 1% nonidet P-40, 0.5% sodium deoxycholate, and 0.1% SDS, pH 8.0). The total protein concentrations were determined using the bicinchoninic acid (BCA) protein assay (Thermo Fisher Scientific, Waltham, MA, USA). 40 μg of total cellular proteins were loaded to the gels after denatured in 4× Laemmli buffer (Bio-Rad, Hercules, CA, USA) at 37 °C for 1 h. Proteins were separated by sodium dodecyl sulfate–polyacrylamide gel electrophoresis (SDS-PAGE) and transferred to polyvinylidene fluoride (PVDF) membranes (IPVH0010, Millipore, Darmstadt, Germany). The PVDF membrane was treated with primary and secondary antibodies, and then, proteins were visualized using Clarity ECL substrate solution (Bio-Rad, Hercules, CA, USA) by Fusion-FX7 (Vilber Lourmat, Paris, France). β-Actin (A5316, Sigma-Aldrich, St. Louis, MO, USA) and GAPDH (5174, Cell Signaling Technology, Denver, MA, USA) were used as loading controls. The primary antibody for SVIP (HPA039807) was purchased from Sigma-Aldrich. The primary antibodies against gp78 (9590), Hrd1 (14773), Derlin1 (8897), ERα (8644), and p53 (2524) were purchased from Cell Signaling Technology (Denver, MA, USA), and p97/VCP (612182) were obtained from BD Transduction Laboratories (San Jose, CA, USA). The secondary antibodies (Goat anti-rabbit-31460 and Goat anti-mouse-31430) were purchased from Thermo Fisher Scientific (Waltham, MA, USA). All Western blot experiments were performed at least in three independent replicates.

### 2.8. Cell Proliferation and Migration

The proliferation rate of SVIP siRNA-transfected and negative siRNA-transfected MCF-7, T47D, ZR-75-1, and SK-BR-3 cells was monitored using the xCELLigence impedance-based real-time cell analysis system (ACEA Biosciences, San Diego, CA, USA). MCF-7, T47D, and SK-BR-3 cells (8000 cells/well) and ZR-75-1 cells (10,000 cells/well) were seeded on an E-plate-16 at the optimal cell density, and cell proliferation was monitored every 30 min. The electrical impedance measured by the RTCA software and data were represented as cell index. Impedance is correlated with an increase in the number of cells on the well by measuring the cell index.

The wound healing assay SVIP siRNA-transfected and negative siRNA transfected MCF-7 and T47D cells were plated at 2 × 10^5^ cells per well into a Cytoselect wound-healing assay 24-well plate (Cell Biolab, San Diego, CA, USA). Each well contained an insert that created a wound field. After 24 h, the insert was gently removed, creating a gap of 0.9 mm. The migration of the cells into the wound field was monitored for 72 h, and images were taken with the microscope (Olympus CKX41). The analysis of wound closure % was determined by using the ImageJ software (http://imagej.nih.gov/ij/, National Institutes of Health, Bethesda, MD, USA).

The migratory ability of MCF-7 and T47D cells transfected with SVIP siRNA was assessed using 24-well Transwell inserts (8 µm pore size; Greiner Bio-one, Kremsmünster, Austria). A total of 1 × 10^4^ cells were plated in the upper chambers of Transwell filters in 100 µL phenol red-free DMEM with 5% DCT-FBS. The cell migration was stimulated through the membranes by DMEM (650 µL) containing 20% FBS in the lower chambers as a chemoattractant. After 48 h of incubation at 37 °C, the migratory cells on the lower membrane surface of the insert were fixed with methanol at room temperature and stained with 0.05% crystal violet solution (Serva, Heidelberg, Germany). Migration was quantified by counting stained cells in five randomly selected fields using the microscope, and the data were expressed as the mean percentage of migrated cells compared with the vehicle control groups.

### 2.9. Statistical Analysis

The significant differences in SVIP expression levels in the different cancer tissues were analyzed by the Mann–Whitney U test in the TNMplot database (* *p <* 0.01). Then, distributions of SVIP gene expression levels in the different cancer type was computed by the Wilcoxon test in the TIMER database (* *p <* 0.05; ** *p <* 0.01; *** *p <* 0.001). The tumor, normal, and metastatic breast cancer data were compared using the Kruskal–Wallis test in the TNM plot database. The statistical significance cutoff was set at *p* < 0.01. In Kaplan–Meier Plotter database, the survival rate with *p* values was analyzed by the log rank test. *p* < 0.05 was considered statistically significant. The cutoff value of SVIP gene expression was chosen as the median. The dataset was split into two groups of patients, and plots were generated accordingly. Welch’s *t*-test and Dunnett–Tukey–Kramer’s tests estimated the significance of SVIP expression levels based on clinicopathological features in bc-GenExMiner v4.8 (*p <* 0.001). The differences in SVIP promoter methylation levels between breast cancer and normal tissue were analyzed by Welch’s *t*-test in the UALCAN database. *p* < 0.05 was considered significant. Data are presented as means ± standard deviation (SD). For experiments comparing differences between groups were performed by Student’s *t*-test using GraphPad Prism software. The significance threshold was accepted as *p* < 0.05. The correlation between SVIP expression and methylation was evaluated using Spearman’s and Pearson’s correlation analyses and statistical significance in cBioPortal. The correlation of gene expression was evaluated using Spearman’s correlation and the p-value in TIMER. *p* < 0.05 was considered statistically significant if not especially noted.

## 3. Results

### 3.1. The mRNA Expression of SVIP in Different Cancer Types

To investigate the possible differential expression of *SVIP* in tumors and normal tissues, the gene expression of *SVIP* was compared in normal and tumor tissues using the TNMplot database. The expression of *SVIP* mRNA in tumor tissue was significantly higher than in normal tissue in most cancer types, including acute myeloid leukemia, breast, colon, liver, pancreas, thyroid, and prostate cancers (Figure 1A). Interestingly, *SVIP* expression was higher in renal chromophobe cell carcinoma (Renal CH) and lung adenocarcinoma (Lung AC) while lower in renal clear cell carcinoma (Renal CC), renal papillary cell carcinoma (Renal PA), and lung squamous cell carcinoma (Lung SC) than in normal tissue (Figure 1A). When the TIMER database with RNA-seq data of multiple malignancies in TCGA was used to verify the expression of *SVIP* in pan-cancer, the analysis revealed that breast-invasive carcinoma (BRCA), cholangiocarcinoma (CHOL), colon adenocarcinoma (COAD), liver hepatocellular carcinoma (LIHC), and lung adenocarcinoma (LUAD) had higher *SVIP* expression; glioblastoma multiforme (GBM), head and neck squamous cell carcinoma (HNSC), kidney renal clear cell carcinoma (KIRC), kidney renal papillary cell carcinoma (KIRP), and lung squamous cell carcinoma (LUSC) had lower expression of *SVIP* than in normal tissue. Importantly, the expression of *SVIP* was significantly upregulated in all the subtypes of breast cancer, namely luminal A, luminal B, HER2 positive, and triple-negative (basal-like) types (Figure 1B). Taken together, these results demonstrated that the *SVIP* mRNA expression was differentially regulated in most cancer types compared with normal tissues, particularly in breast cancer (Figure 1A,B and Appendix A).

### 3.2. Promoter Methylation Level and Genomic Alteration Analysis of SVIP in Breast Cancer

To investigate the possible relationship between *SVIP* expression and DNA methylation, the promoter methylation level of the *SVIP* in breast cancer tissues was analyzed using the UALCAN tool from the TCGA dataset. The level of methylation on the *SVIP* promoter was significantly lower in breast cancer compared to the normal tissues, as shown in Figure 2A (*p* = 3.02 × 10^−4^). Similarly, when the cBioPortal database was used, *SVIP* mRNA expression showed a negative correlation with *SVIP* methylation, although not very strong (Spearman = −0.19, *p* = 8.81 × 10^−7^; Pearson = −0.10, *p* = 0.0128) (Appendix A).

We then explored the potential genetic alterations of *SVIP* in the context of breast cancer using the c-BioPortal online tool. The OncoPrint results showed that in 142 of 1084 breast cancer patients (13%), the *SVIP* gene had altered either amplification, deep deletion, high mRNA expression, or low mRNA expression; the most common alteration type was found to be high mRNA expression (Figure 2B). When alteration frequency was assessed for different breast carcinoma samples, namely breast invasive carcinoma, breast invasive ductal carcinoma, breast invasive lobular carcinoma, and breast invasive mixed mucinous carcinoma, the frequency of *SVIP* gene alteration rates varied from 5.88% to 18.18%, where mRNA high alteration was the most common one, with the ratios varying from 5.88% to 11.69% (Figure 2C).

### 3.3. Associations between SVIP Expression Levels and the Clinicopathological Parameters and Its Prognostic Value in Breast Cancer Patients

As *SVIP* mRNA is overexpressed in primary breast tumors compared to normal tissues, we next analyzed the expression of *SVIP* in distinct clinicopathological stages, parameters, and intrinsic subtypes of breast cancer. Firstly, *SVIP* mRNA was found to be highly expressed in every clinicopathological stage of breast cancer compared with normal tissues using the UALCAN portal (Figure 3A). To further evaluate the relationship between *SVIP* expression levels and various clinicopathological parameters, the bc-GenExMiner (v4.8) online tool was utilized. *SVIP* mRNA was found to be highly expressed not only in the breast cancer tissues but also in the tumor-adjacent tissue compared to normal tissues, which may suggest that *SVIP* may also be expressed in stromal cells (Figure 3B). Intriguingly, *SVIP* mRNA levels were higher in patients with ER-positive than ER-negative tumors, with PR-positive than PR-negative tumors, and with p53 wt compared to p53 mutant tumors (Appendix A). On the other hand, *SVIP* mRNA levels were significantly lower in the HER2-positive group than in the HER2-negative group (Appendix A).

When the RNA-seq data of *SVIP* expression in normal, tumor, and metastatic breast tissues was evaluated via the TNMplot database, the analysis revealed a higher expression level of *SVIP* in tumor samples not only than in normal samples, but also metastatic samples (Figure 3C). Interestingly, metastatic samples have lower *SVIP* expression compared to normal samples (*p =* 1.06 × 10^−5^). However, as a subanalysis of metastatic tumors involving ER and HER2 status is not available, it is not certain that low *SVIP* expression in metastatic tissue is biologically relevant.

The Kaplan–Meier Plotter database was used to investigate the prognostic potential of *SVIP* in breast cancer patients. The RNA-seq data in Kaplan–Meier Plotter database is mainly extracted from Gene Expression Omnibus (GEO), European Genome-phenome Atlas (EGA), and TCGA. All results are displayed with *p*-values from a long-rank test. Results revealed that lower *SVIP* mRNA expression levels were associated with a worse prognosis of overall survival (OS) (HR = 0.74, *p* = 0.026) (Figure 3D) and distant metastasis-free survival (DMFS) (HR = 0.61, *p* = 0.00026) (Figure 3E) in breast cancer. Furthermore, when the effect of *SVIP* expression on the overall survival of breast cancer patients was analyzed considering the breast cancer subtype, lower *SVIP* mRNA expression levels were associated with a worse prognosis in the luminal A, luminal B, and TNBC subtypes (Figure 3F).

Considering that previously published reports suggesting that the tumor suppressor role of SVIP and mRNA expression and survival analysis have not exhibited a consistent pattern, the analysis of SVIP protein expression was next performed. Strikingly, SVIP protein expression was much lower in total and all subtypes of breast cancer samples compared to normal tissues (Figure 4A,B).

### 3.4. Expression of SVIP and Related Key ERAD Proteins in Breast Cancer Cell Lines

As our analysis revealed that mRNA and protein expression levels of SVIP were different in breast tumor tissues compared to control tissues, we examined the expression level of SVIP protein in commercially available breast cancer cell lines (Figure 5A). For this aim, SVIP protein expression levels were analyzed in the cell lines that represent models for luminal A [MCF-7 (ER+, PR+, HER2−), T47D (ER+, PR+, HER2−), and ZR75-1(ER+, PR+/−, HER2)], luminal B [BT-474 (ER+, PR+, HER2+)], HER2+ [SKBR3 (ER−, PR−, HER2+)], triple-negative breast cancer [MDA-MB-231 (ER−, PR−, HER2−)] breast cancer subtypes and non-tumorigenic breast epithelial cell line [MCF-10A (ER−, PR−, HER2−)] [27,28]. SVIP protein expression was significantly higher in breast cancer cell lines compared to non-tumoral breast epithelial cell lines (Figure 5A). We have previously reported androgen-dependent regulation of SVIP in prostate cancer [16]. Therefore, next, the estrogen dependency of SVIP expression in ER+ breast cancer cells was evaluated using 17β-estradiol (E2), and our results revealed that E2 did not cause any significant alterations in SVIP expression levels in MCF-7, T47D, and ZR75-1 cells (Figure 5B).

SVIP, the first identified endogenous ERAD inhibitor, inhibits gp78-mediated ERAD by competing with p97/VCP and Derlin1 [13]. Given that SVIP expression is high in breast cancer cells, we evaluated the other functional key players of gp78-mediated ERAD. The expression of p97/VCP, the key protein functioning on the retrotranslocation step of ERAD, was significantly higher in luminal B subtype BT-474 cells but low in HER2+ subtype SKBR3 cells. gp78 and Derlin1 expression were higher in MCF-7 and BT474 cells (Figure 5C). Our results indicated that breast cancer cell lines present significantly higher SVIP expression compared to non-tumorigenic epithelial MCF-10A cell lines. On the other hand, the expression of other major proteins functioning in gp78-mediated ERAD was high only in MCF-7 and BT474 cells. Interestingly, the Hrd1 level, the other major endoplasmic reticulum resident ubiquitin ligase other than gp78, displayed an expression pattern similar to SVIP expression.

### 3.5. The Effect of SVIP in the Proliferation and Migration of Breast Cancer Cell Lines

Given the potential for the SVIP gene to be effective in breast cancer progression, functional analysis was performed in the next step. Firstly, the effect of SVIP on the proliferation of MCF-7, ZR-75-1, T47D, and SK-BR-3 cells was determined with the xCELLigence Real-Time Cell Analyzer. Silencing of SVIP expression using siRNA oligonucleotides against SVIP mRNA enhanced the proliferation of MCF-7 and ZR-75-1, but not T47D and SK-BR-3 cells (Figure 6A–D). Interestingly, while silencing of SVIP expression significantly decreased p53 protein levels in MCF-7 and ZR-75-1 cells, there was no change in p53 mutant T47D and SK-BR-3 cells.

As Hrd1 was reported as the ubiquitin ligase for p53 and target p53 for proteasomal degradation [29], we have simultaneously silenced SVIP and Hrd1 expression to assess whether SVIP increases p53 protein levels by inhibiting Hrd1-mediated p53 degradation. Our results suggested that SVIP knockdown was no longer able to downregulate p53 expression in Hrd1-silenced MCF7 cells (Figure 6E).

Next, the role of SVIP silencing on the motility of MCF-7 and T47D was tested by an in vitro wound healing model using Cytoselect 24-well plate, which provides a standardized wound area via proprietary treated inserts. Our data indicated that silencing of SVIP increased the rate of wound closure of both MCF-7 and T47D compared to that observed in negative siRNA-transfected control cells (*p < 0.05* for siSVIP MCF7, *p* < 0.05 for siSVIP T47D) (Figure 7A,B). Using the Transwell Boyden chamber to analyze the migration ability consistently revealed that silencing of SVIP increased the migration of both MCF-7 and T47D cell lines (*p < 0.05* for siSVIP MCF-7, *p < 0.05* for siSVIP T47D) (Figure 7C,D). To conclude, our data suggest that SVIP has a function in the regulation of cell proliferation and migration of breast cancer cells.

## 4. Discussion

The endoplasmic reticulum is involved in multiple cellular processes, including protein and lipid biosynthesis, protein folding and transport, and calcium homeostasis. Nutrient deficiency, oxidative stress, high metabolic demand, and calcium imbalance in the tumor microenvironment disrupt endoplasmic reticulum homeostasis and cause excessive accumulation of misfolded/unfolded proteins, with resultant endoplasmic reticulum stress. The major degradation pathway that the endoplasmic reticulum uses is the ERAD pathway facilitating proteasome-dependent degradation of unfolded or misfolded proteins4. ERAD is a multi-component and complex system comprising many proteins, including gp78, Sel1L, Hrd1, Derlin-1/2 (Derl1/2), p97/VCP functioning on translocation, ubiquitination, and proteasomal degradation of non-native proteins. Besides functioning as a downstream response to UPR, recent evidence showed that the ERAD complex has UPR-independent functions, and there is a crosstalk between UPR and ERAD [30].

The first identified endogenous ERAD inhibitor SVIP is a multifunctional protein that has also been associated with autophagy, lysosomal dynamic stability, VLDL (very low-density lipoprotein) trafficking, and secretion [31,32,33]. Furthermore, its gradually increasing expression in the developing nervous tissues, together with structural studies, have been implicated that SVIP is a novel compact myelin protein [34,35]. Very recently, SVIP was reported to be expressed in the adrenal gland at a consistently high level during postnatal development, suggesting that SVIP may function throughout the developmental process of the adrenal gland. Hence, modulation of SVIP expression was reported to alter not only the expression levels of steroidogenic genes and hormone output in adrenal cortex cells but also the expression of genes required for de novo cholesterol biosynthesis, uptake, and trafficking [35].

SVIP has recently started to attract attention in cancer biology studies. It has been reported that SVIP undergoes DNA hypermethylation-associated silencing in some cancer cells, especially in head and neck cancer. SVIP exhibited tumor suppressor features and its inhibition in SVIP-hypermethylated cancer cells. Interestingly, except for head and neck (50%), esophageal (23%), cervical (21%) cancers, and hematological malignancies (14%), particularly B cell lymphoma (33%), the *SVIP* promoter CpG island was most often found to be unmethylated in the other cancer types including prostate and breast cancers [18]. SVIP has also been identified as an androgen-regulated gene in prostate cancer and glioma [16,17]. It has been proven that SVIP is downregulated, but other ERAD components and androgen receptors (AR) are upregulated in glioma and androgen-dependent prostate cancer cell lines with R1881 treatment. Furthermore, the regulation of the levels of SVIP and ERAD components leads to enhanced ERAD proteolytic activity, which was found to be related to prostate tumorigenesis [16]. Similarly, the decreased SVIP expression, as well as increased AR expression, in glioma tissues correlated with gliomas progressing from low to high grades. Interestingly, the suppression of SVIP by AR was associated with decreased p53 expression, and overexpression of SVIP increased cell death only in p53wt glioma cell lines, suggesting the downregulation of SVIP [17].

Apart from prostate cancer, breast cancer is another hormone-dependent cancer with a high incidence. Hormone receptor-positive breast and prostate cancers share several similarities, one of which is their dependence on the respective male and female hormones for their continued growth. While estrogen and androgen promote the growth of these cancers, their deficiency results in low differentiation, increased proliferation rate, and unresponsiveness to the treatment [36]. As SVIP was identified as an androgen-dependent gene and reported to be associated with tumorigenesis of both prostate cancer and glioma, we have investigated the SVIP expression in various cancers, with a particular focus on breast cancer due to its estrogen dependency.

First, the differential *SVIP* mRNA expression was observed in most human cancers through multi-omics data analysis. Our results revealed that *SVIP* mRNA expression was upregulated in many cancer types, including adrenal, breast, and prostate cancers (Figure 1A). Importantly, our data showed that *SVIP* mRNA expression was increased in all four intrinsic molecular subtypes of breast cancer (HER2+, triple negative (basal like), luminal A, and luminal B) compared to normal tissues using the GEPIA database (Appendix A). On the other hand, *SVIP* mRNA expression was significantly decreased, especially in head and neck squamous cell carcinoma (Figure 1B). These results are in line with a study by Llinàs-Arias et al. suggesting *SVIP* expression is upregulated in breast cancer, prostate cancer, lung cancer, and skin cancer while significantly decreased in head and neck tumors [18].

Epigenetic modifications, such as DNA methylation, have a significant effect on tumorigenesis development and malignant transformation [37]. DNA methylation plays an important role in the progression and prognosis of breast cancer patients. Also, it has an important regulatory role in gene expression in cancer cells [38,39]. Previously, the downregulation of *SVIP* in some cancers, such as head and neck tumors, was shown to be associated with hypermethylation of *SVIP* promoter CpG island [18]. Here, epigenetic-associated transcriptional regulation of *SVIP* was investigated in breast tumors, and the promoter methylation level of *SVIP* in breast cancer was found to be significantly lower than in normal tissues, and *SVIP* mRNA expression was negatively correlated with *SVIP* methylation (Figure 1A and Figure 2A). Consistently, the most common type of genomic alteration was high mRNA expression in all breast cancer types (Figure 2B).

Analysis of SVIP protein levels in breast cancer tissues is critical, given that the epigenetic-associated transcriptional silencing of SVIP has been previously linked to its tumor suppressor function in some tumors [18], and our results suggested low promoter methylation and high mRNA expression of SVIP in breast tumor tissues. Importantly, our data revealed that SVIP protein levels in breast cancer tissues compared to normal tissues are low despite the increased mRNA levels (Figure 1A,B and Figure 4A). Gene expression variation at the mRNA level is not necessarily consistent with the protein level, and only 50% or less of variation in protein levels is explained by variations in mRNA levels for many organisms [39]. Although genes that are particularly specified for breast cancer, such as ESR1, PGR, HER2, EGFR, and AR are highly correlated in mRNA-protein expression, nearly 50% of gene expressions are not consistent with their protein levels both in tumors and cell lines [40]. There are several reasons for this situation that may not be mutually exclusive. First, there are many complicated and varied post-transcriptional mechanisms in turning mRNA into proteins. Second, in vivo half-lives of proteins can differ significantly.

Another major outcome of our study was that we identified differences in SVIP protein expression levels between cell lines and in tumors in breast cancer. Our immunoblotting data confirmed that SVIP protein levels are significantly higher in breast cancer cell lines (MCF-7, T47D, ZR75-1, BT-474, SK-BR-3, and MDA-MB-231) compared to human mammary epithelial cells (MCF-10A) (Figure 5A). As cell lines have more genetic aberration than primary tumors, they exhibit a remarkable increase in lineage-restricted profiles that fail to fully represent the intratumoral heterogeneity of individual breast tumors, even though, as a group, they indeed represent the heterogeneity of human breast tumors [41]. Consistently, a previous study that comprehensively compared the molecular portraits between cell lines and tumors in breast cancer reported the similarity and dissimilarity of molecular features between breast cancer tumors and cell lines and suggested that cell lines and tumors show high gene expression-based correlations, but the correlations in protein expression levels are low [40]. The composition of the cell culture medium, the effect of immortalization processes, and/or the absence of multicellular interfaces with which tumor cells in vivo for cells grown in cell culture plates are only some potential interpretations [42].

It has been previously reported that SVIP restoration in deficient head and neck tumor cells blocks ERAD consisting of its role as an ERAD inhibitor [18]. Furthermore, synthetic androgen treatment downregulated SVIP levels but upregulated other ERAD components, including gp78, p97/VCP, and Derlin1, enhancing ERAD proteolytic activity both in glioma and prostate cancer cells [16,17]. When the expression of key gp78-mediated ERAD pathway was evaluated in a breast cancer cell panel, we found that p97/VCP, Derlin1, and gp78 expression was higher only in MCF-7 and BT474 cells. On the other hand, the main ERAD ubiquitin ligase Hrd1 had an expression pattern similar to SVIP. This may be due to the regulatory effect of SVIP and Hrd1 on either the expression level or functionality of gp78. While SVIP inhibits gp78-mediated ERAD through the removal of the p97/VCP and Derlin1 from the gp78 complex, Hrd1 has been shown to target gp78 for proteasomal degradation independent of the ubiquitin ligase activity of gp78 [13,43,44]. Another possible reason may be the enhanced endoplasmic reticulum stress conditions in breast cancer and the upregulation of Hrd1 and SVIP expressions by endoplasmic reticulum stress [13,45]. Previously, it has been reported that p53 wild-type cell lines may be responsive to SVIP depletion by siRNA, thereby leading to decreased p53 expression and increased cell proliferation of glioma cells; however, p53 mutant cell lines may not [17]. Given that SVIP silencing resulted in diminished p53 expression in p53 wt MCF7 cells and Hrd1 was reported as the ubiquitin ligase for p53 and target p53 for proteasomal degradation [29], simultaneous knocking down of SVIP and Hrd1 was performed. SVIP knockdown did not result in the downregulation of p53 in Hrd1-silenced MCF7 cells, suggesting that SVIP increases p53 protein levels by inhibiting Hrd1-mediated p53 degradation (Figure 6E). Notably, silencing of SVIP in MCF7 cells increased the levels of Hrd1, further indicating the possible regulatory role of SVIP on Hrd1-mediated p53 degradation, the mechanism of which requires further investigation.

Although SVIP silencing resulted in diminished expression of p53 together with increased proliferation rates in p53 wt MCF7 and ZR-75-1 cells but not in p53 mutant T47D and SK-BR-3 cells, the migration of MCF-7 and T47D cell lines was affected by SVIP silencing, suggesting that breast cancer cells with lower SVIP expression might have higher migration ability (Figure 7A–D). Interestingly, we found that SVIP expression is higher in primary breast tumors and lower in metastatic breast tumors compared to normal tissues (Figure 3C). Consistent with these findings, breast cancer patients with lower SVIP expression exhibited a lower probability of survival compared to the patients with overexpressed SVIP (Figure 3D–F).

It is important to highlight that in silico analysis can have considerable limitations, such as most platforms do not allow further subanalysis for cellular heterogeneity and often have small cohorts. Although several questions are still remaining, this manuscript sheds some light on the role of SVIP in breast tumorigenesis and indicates that SVIP has tumor-suppressor-like properties in breast cancer cells. The higher mRNA expression and lower protein expression of SVIP in breast tumor tissues compared to normal tissues may indicate a different regulatory pattern for SVIP, which deserves further investigation. Moreover, future studies on the potential crosstalk between Hrd1 and SVIP, particularly in the context of regulating p53 protein levels, will be highly promising for clinical translation.

## Figures and Tables

**Figure 1 cells-12-01362-f001:**
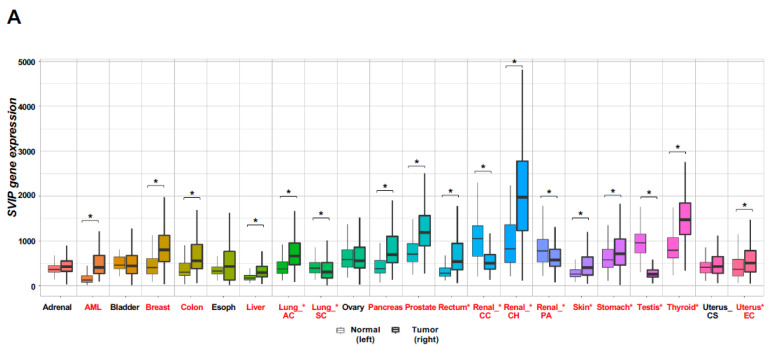
*SVIP* expression levels in different human tumors. (**A**) Increased or decreased expression of *SVIP* in different tumors compared to normal tissues via the TNMplot database. Significant differences by Mann–Whitney U-test are marked with red color (* *p* < 0.01). (**B**) Human *SVIP* expression levels of different tumor types from the TCGA database were investigated by TIMER (* *p* < 0.05, ** *p* < 0.01, *** *p* < 0.001). Abbreviations: ACC: adrenocortical carcinoma; BLCA: bladder urothelial carcinoma; BRCA: breast invasive carcinoma; CESC: cervical squamous cell carcinoma; CHOL: cholangiocarcinoma; COAD: colon adenocarcinoma; DLBC: lymphoid neoplasm diffuse large B cell lymphoma; ESCA: esophageal carcinoma; GBM: glioblastoma multiforme; LGG: brain lower grade glioma; HNSC: head and neck squamous cell carcinoma; KICH: kidney chromophobe; KIRC: kidney renal clear cell carcinoma; KIRP: kidney renal papillary cell carcinoma; LAML: acute myeloid leukemia; LIHC: liver hepatocellular carcinoma; LUAD: lung adenocarcinoma; LUSC: lung squamous cell carcinoma; MESO: mesothelioma; OV: ovarian serous cystadenocarcinoma; PAAD: pancreatic adenocarcinoma; PCPG: pheochromocytoma and paraganglioma; PRAD: prostate adenocarcinoma; READ: rectum adenocarcinoma; SARC: sarcoma; SKCM: skin cutaneous melanoma; STAD: stomach adenocarcinoma; TGCT: testicular germ cell tumors; THCA: thyroid carcinoma; THYM: thymoma; UCEC: uterine corpus endometrial carcinoma; UCS: uterine carcinosarcoma; and UVM: uveal melanoma; AML: acute myeloid leukemia; Lung_AC: lung adenocarcinoma; Lung_SC: lung squamous cell carcinoma; Renal_CC: renal clear cell carcinoma; Renal_CH: renal chromophobe cell carcinoma; Renal_PA: renal papillary cell carcinoma; Uterus_CS: uterine carcinosarcoma; Uterus_EC: uterine corpus endometrial carcinoma.

**Figure 2 cells-12-01362-f002:**
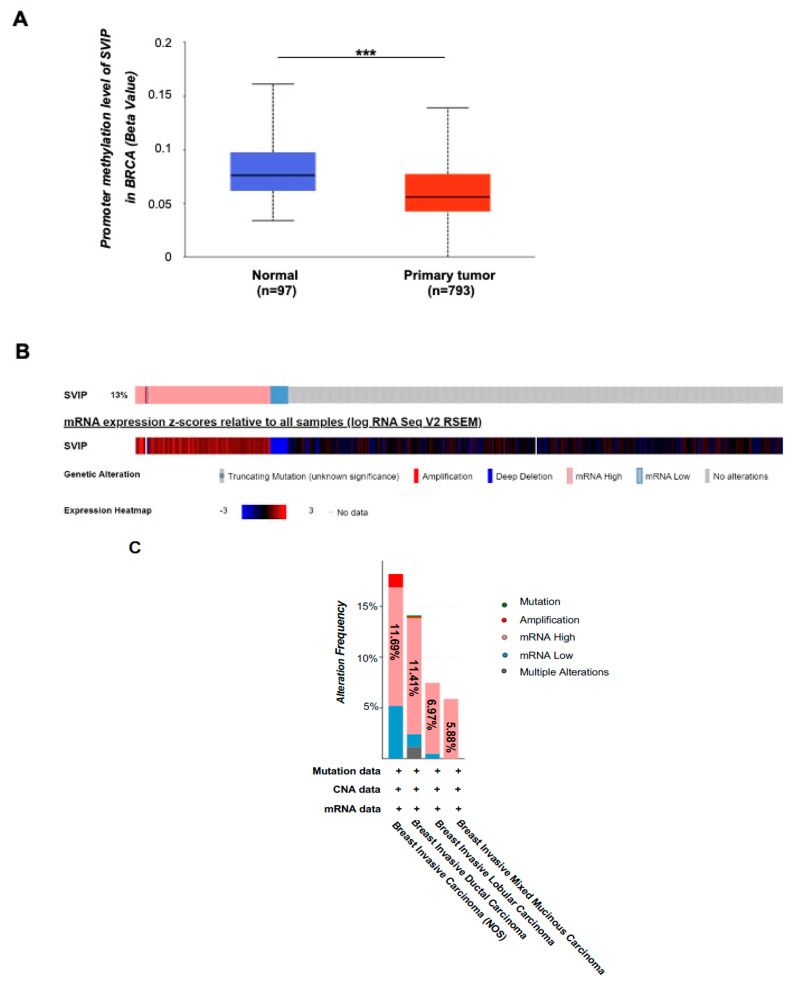
Promoter methylation level and genomic alterations of *SVIP*. (**A**) The promoter methylation level of *SVIP* in normal and breast cancer tissues (UALCAN) (*p* = 3.02 × 10^−4^) ***, *p* < 0.001. (**B**) Oncoprint of *SVIP* alteration in breast cancer (c-BioPortal). (**C**) *SVIP* gene expression and mutation analysis in different breast cancer types (c-BioPortal).

**Figure 3 cells-12-01362-f003:**
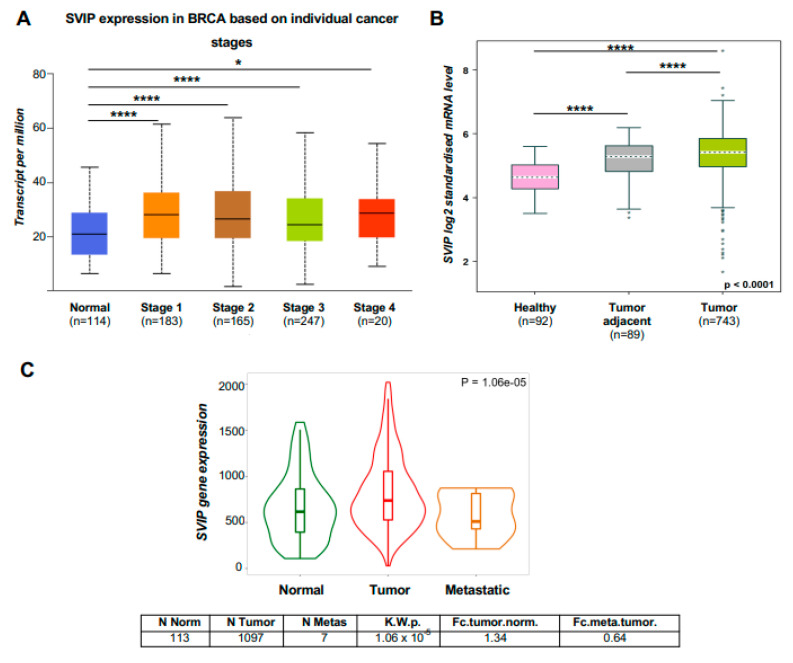
UALCAN, TNMplot, and bc-GenExMiner v4.8 portal analyses of breast cancer samples from the TCGA and GTEx datasets. (**A**) Expression of *SVIP* in different stages of breast cancer (UALCAN) (* *p* < 0.05; ****, *p* < 0.0001). (**B**) *SVIP* expression in breast cancer, tumor-adjacent normal tissues and healthy tissues (**** *p* < 0.0001) (bc-GenExMiner software). (**C**) RNA-seq data of *SVIP* expression in normal, tumor, and metastatic tissues by the TNMplot database. (**D**,**E**) Overall survival (OS) and distant metastasis-free survival (DMFS) curves of BRCA by different expression levels of *SVIP* in the Kaplan–Meier Plotter database (*p* < 0.05). (**F**) Survival probability between breast cancer patients with molecular subtypes and *SVIP* expression using Kaplan–Meier Plotter database (*p* < 0.05).

**Figure 4 cells-12-01362-f004:**
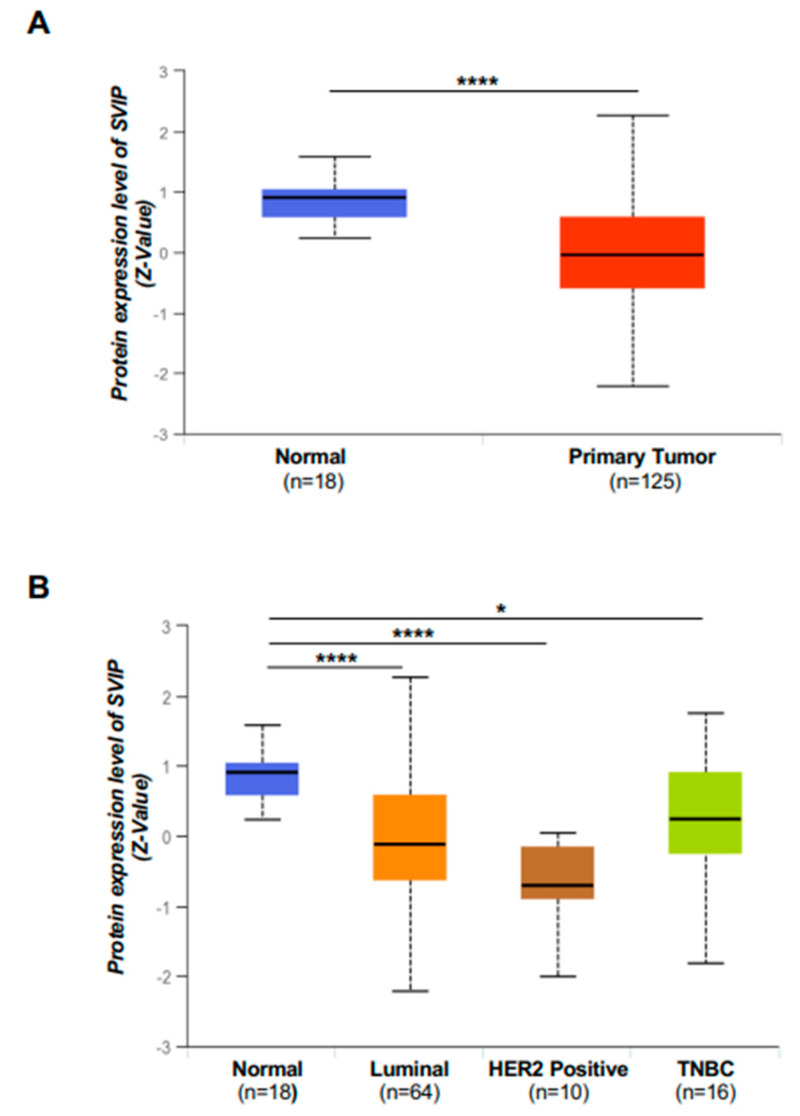
The protein level of SVIP in breast cancer tissue. (**A**) Protein expression of SVIP in breast cancer tumors and normal tissues detected by UALCAN database (*p* < 0.05). (**B**) Protein expression of SVIP in breast cancer patients with molecular subtypes (UALCAN) (* *p* < 0.05; **** *p* < 0.0001).

**Figure 5 cells-12-01362-f005:**
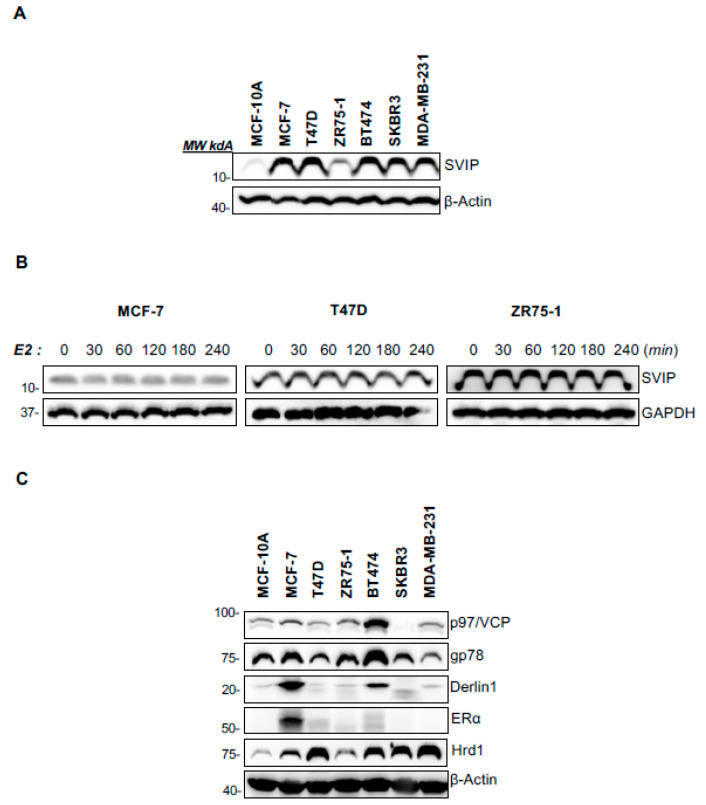
Expression of SVIP level in different breast cancer cell lines. (**A**) SVIP expression in breast cancer cell lines and non-tumorigenic breast epithelial cell lines by Western blot analysis. Western blot is representative of three replicates with similar results. (**B**) SVIP protein levels in breast cancer cell lines (MCF-7, T47D, ZR-75-1) were determined by Western blot analysis. β-actin or GAPDH was used as a loading control (n = 3). (**C**) Expression of ERAD components in different breast cancer cell lines (n = 4).

**Figure 6 cells-12-01362-f006:**
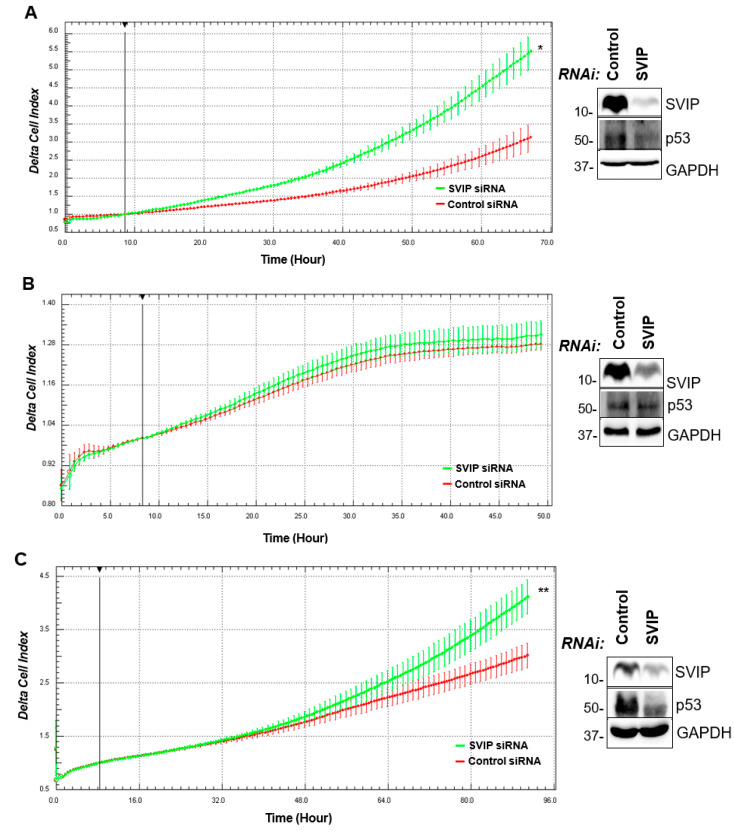
Effect of SVIP silencing on breast cancer tumorigenesis. (**A**) Real-time proliferation analysis of MCF-7, (**B**) T47D cells, (**C**) ZR-75-1 cells, and (**D**) SK-BR-3 cells with xCELLigence RTCA. Cells were monitored for the indicated time. A Representative Cell Index profile was presented for one biological replicate (n = 3) with five technical replicates (* *p* = 0.00003, ** *p* = 0.001). Cells were transfected with siRNA oligonucleotides (SVIP and control siRNA). Expression of SVIP and p53 was evaluated by immunoblotting using GAPDH as the loading control. Error bars are presented as standard deviations. (**E**) Effect of simultaneous knockdown of SVIP and Hrd1 on p53 levels. MCF-7 cells were transfected with siRNA oligonucleotides (SVIP, Hrd1, and control siRNA) and 72 h later subjected to immunoblotting analysis of SVIP, Hrd1, p53, and GAPDH (n = 3).

**Figure 7 cells-12-01362-f007:**
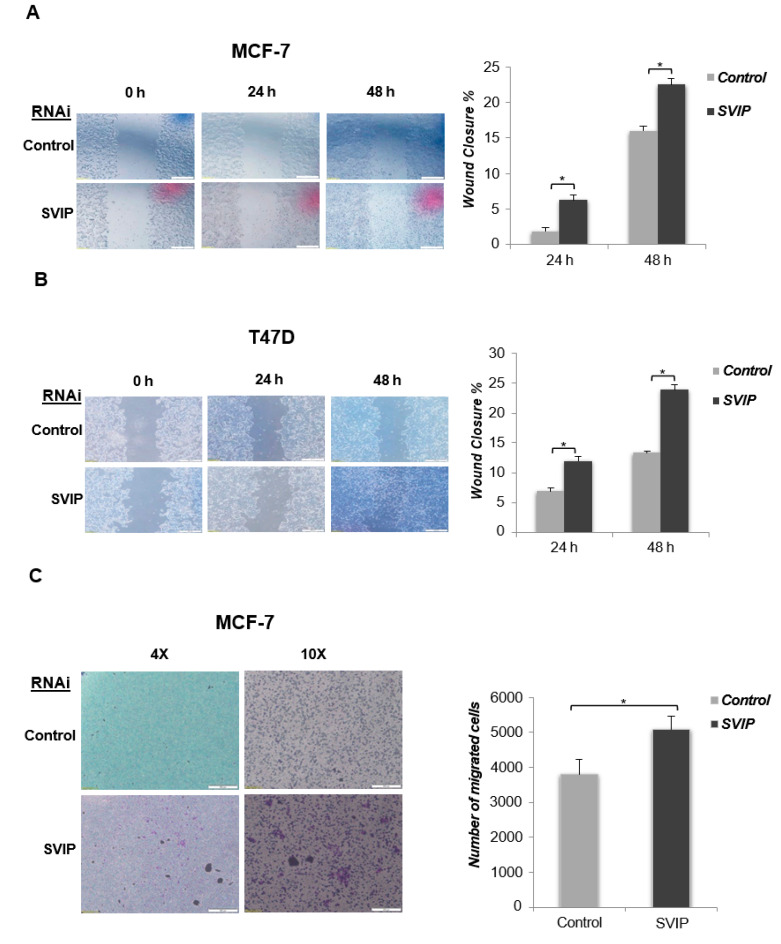
The role of SVIP in breast cancer migration. Wound healing assay was performed in 0, 24, and 48 h in transfected (**A**) MCF-7 (*p* = 1.032 × 10^−6^; *p* = 8.81 × 10^−7^) and (**B**) T47D cells (*p* = 1.74 × 10^−6^; *p* = 3.68 × 10^−9^). The analysis of wound closure % was determined using the ImageJ software. Two independent biological and three technical repeats per experiment were used. *p*-values were calculated with respect to control siRNA transfected cells by Student’s *t*-test (* *p* < 0.05). (**C**) Transwell migration of MCF-7 (*p* = 0.0005) and (**D**) T47D (*p* = 0.001) cell lines. Cells were allowed to migrate for 72 h. Pictures were taken at ×4 and ×10 magnification. Three independent experiments were analyzed with the Student’s *t*-test (* *p* < 0.05). Error bars are presented as standard deviations.

## Data Availability

Data will be made available on request.

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
