# Peer review of "Differential Expression and Function of SVIP in Breast Cancer Cell Lines and In Silico Analysis of Its Expression and Prognostic Potential in Human Breast Cancer"

_cells, 2023, doi:10.3390/cells12101362_

Round 1

Reviewer 1 Report

In this manuscript, Sahar EA and Kirmizibayrak PB report their study on SVIP expression and its role in cancer. SVIP was previously reported to be inhibitor of ERAD by competing with gp78 to bind to p97/VCP. In rent year, SVIP has been implicated in progression of several cancer types, including glioma, prostate, and head and neck cancers. This study extended the previous reports by systematically analyzing the data of several RNA-seq and gene array studies to evaluate the SVIP gene expression in a variety of cancers, with an emphasis on breast cancer. They found that SVIP was significantly overexpressed in at least 12 different types of cancers. Overexpression of SVIP was also demonstrated in various breast cancer cell lines, whereas the expression of the key proteins of gp78 ERAD complex did not exhibit similar pattern. They also found the tumor suppressor protein p53 as a down-stream mediator of SVIP function in breast cancer, which is similar to what has been reported in glioma. But this study added new information that SVIP regulates cancer cell migration irrespective of the functional status of p53 in breast cancer cell lines. They also showed that breast cancer patients with lower SVIP expression correlate with a lower probability of survival. This study is well designed and clearly written. The results are convincing. It differs from previous report on SVIP in a specific type of cancer by performing a systematic analysis in a variety of cancers. It would make the study more interesting if the author could provide some mechanistic studies as to how SVIP regulates p53 protein levels and cancer cell migration. Here are my concerns:

1.    The labels in most of the figures showing bioinformatic analysis are too small and used font with variable sizes. Please correct.

2.    Although the authors described there as not ERAD complex exhibiting expression pattern similar to that of SVIP in breast cancer cell lines, Hrd1 expression in fact clearly showed an increase (Fig. 4C). This should be described in the text.

3.    How SVIP regulates the levels of p53 is unknown. Since it has been previously reported that Hrd1 is an E3 for p53 and target p53 for proteasomal degradation (EMBO J. 2007 Jan 10;26(1):113-22), this reviewer strongly suggests determining whether SVIP increases p53 protein levels by inhibiting Hrd1-mediated p53 degradation. This could be done simply by simultaneously knocking down of SVIP and Hrd1 to see if p53 can still be downregulated. If worked, the results will strengthen the paper.

Author Response

We would like to thank for providing us with the opportunity to revise our manuscript. We would also like to thank you and the reviewers for the time and expertise in providing feedback. We think that all comments raised by the reviewers are legitimate and require consideration. We would like to profoundly thank them for their constructive comments which have greatly improved the manuscript. The manuscript has now been revised extensively in accordance with the comments of the reviewers.Please find below a detailed account of the revisions made to the manuscript.

In accordance with the comments. In the main manuscript the modified and edited parts were highlighted with track change application.

Comment 1: The labels in most of the figures showing bioinformatic analysis are too small and used font with variable sizes. Please correct.

We have revised the manuscript accordingly.

Comment 2: Although the authors described there as not ERAD complex exhibiting expression pattern similar to that of SVIP in breast cancer cell lines, Hrd1 expression in fact clearly showed an increase (Fig. 4C). This should be described in the text.

In the manuscript we have emphasized this data as “Interestingly, the Hrd1 level, the other major endoplasmic reticulum resident ubiquitin ligase other than gp78, displayed an expression pattern similar to SVIP expression.”

Comment 3: How SVIP regulates the levels of p53 is unknown. Since it has been previously reported that Hrd1 is an E3 for p53 and target p53 for proteasomal degradation (EMBO J.2007 Jan 10;26(1):113-22), this reviewer strongly suggests determining whether SVIP increases p53 protein levels by inhibiting Hrd1-mediated p53 degradation. This could be done simply by simultaneously knocking down of SVIP and Hrd1 to see if p53 can still be downregulated. If worked, the results will strengthen the paper.

We deeply thank the reviewer this valuable suggestion. SVIP knockdown was not resulted in downregulation of p53 in Hrd1 silenced MCF7 cells, suggesting that SVIP increases p53 protein levels by inhibiting Hrd1-mediated p53 degradation (Fig S4). Notably, silencing of SVIP in MCF7 cells increased the levels of Hrd1, further indicating the possible regulatory role of SVIP on Hrd1 mediated p53 degradation, the mechanism of which requires further investigation.

Reviewer 2 Report

ER and ER stress-related pathways are of importance in cancer and other diseases. SVIP is an interesting multifunctional protein and a relatively less-known component in ER stress response. Hence, I found the paper very interesting. The manuscript is well-written and clear. The experimental data is sound. The results of this manuscript may increase interest in SVIP.

I suggest the authors look at SVIP protein data available for some of the datasets they have used. Sometimes the transcript and protein levels do not agree, and SVIP in this study has a tumor-suppressing effect. I wonder if the tumors' protein levels are low despite the increased mRNA levels. There are, of course, many mechanisms that may affect protein levels or activity. Heterogeneity is another issue for cancers.

Minor points

For MCF7 wound closure figure, it is difficult to see the cell boundaries or the cells. A different exposure or marking the boundaries may highlight the difference between control and SVIP silenced cells.

Figure 6. Y axis "migtated" should be corrected.

Author Response

We would like to thank for providing us with the opportunity to revise our manuscript. We would also like to thank you and the reviewers for the time and expertise in providing feedback. We think that all comments raised by the reviewers are legitimate and require consideration. We would like to profoundly thank them for their constructive comments which have greatly improved the manuscript. The manuscript has now been revised extensively in accordance with the comments of the reviewers.

Please find below a detailed account of the revisions made to the manuscript.

In the main manuscript the modified and edited parts were highlighted with track change application.

Comment 1 : I suggest the authors look at SVIP protein data available for some of the datasets they have used. Sometimes the transcript and protein levels do not agree, and SVIP in this study has a tumor-suppressing effect. I wonder if the tumors' protein levels are low despite the increased mRNA levels. There are, of course, many mechanisms that may affect protein levels or activity. Heterogeneity is another issue for cancers.

We thank the reviewer for this constructive comment which has greatly improved the manuscript. SVIP protein expression was much lower not only in all clinicopathological stage and subtypes of breast cancer compared to normal tissues using the UALCAN portal (Fig. 4A, B).

Comment 2: For MCF7 wound closure figure, it is difficult to see the cell boundaries or the cells. A different exposure or marking the boundaries may highlight the difference between control and SVIP silenced cells.

We have revised the manuscript accordingly.

Comment 3: Figure 6. Y axis "migtated" should be corrected.

We thank the reviewer for his/her careful review. We have corrected it.